# A Dual-FSM GI LiDAR Imaging Control Method Based on Two-Dimensional Flexible Turntable Composite Axis Tracking

Yu Cao [1,2,3,4], Meilin Xie [1,2,3], Haitao Wang [1,2], Wei Hao [1,2,3], Min Guo [1,2,3], Kai Jiang [1,2], Lei Wang [1,2], Shan Guo [1,2] and Fan Wang [1,2,*]

1   Xi'an Institute of Optics and Precision Mechanics, Chinese Academy of Sciences, Xi'an 710119, China; caoyu@opt.ac.cn (Y.C.); xiemeilin@opt.ac.cn (M.X.); wanghaitao@opt.ac.cn (H.W.); haowei@opt.ac.cn (W.H.); gmnwpu@opt.ac.cn (M.G.); jiangkai@opt.ac.cn (K.J.); wanglei@opt.ac.cn (L.W.); guoshan@opt.ac.cn (S.G.)
2   Key Laboratory of Space Precision Measurement Technology, Chinese Academy of Sciences, Xi'an 710119, China
3   Pilot National Laboratory for Marine Science and Technology, Qingdao 266237, China
4   Collaborative Innovation Center of Extreme Optics, Shanxi University, Taiyuan 030006, China
*   Correspondence: wangfan@opt.ac.cn; Tel.: +86-132-0180-8819

**Abstract:** In this study, a tracking and pointing control system with a dual-FSM (fast steering mirror) two-dimensional flexible turntable composite axis is proposed. It is applied to the target-tracking accuracy control in a GI LiDAR (ghost imaging LiDAR) system. Ghost imaging is a multi-measurement imaging method; the dual-FSM GI LiDAR tracking and pointing imaging control system proposed in this study mainly solves the problems of the high-resolution remote sensing imaging of high-speed moving targets and various nonlinear disturbances when this technology is transformed into practical applications. Addressing the detrimental effects of nonlinear disturbances originating from internal flexible mechanisms and assorted external environmental factors on motion control's velocity, stability, and tracking accuracy, a nonlinear active disturbance rejection control (NLADRC) method based on artificial neural networks is advanced. Additionally, to overcome the limitations imposed by receiving aperture constraints in GI LiDAR systems, a novel optical path design for the dual-FSM GI LiDAR tracking and imaging system is put forth. The implementation of the described methodologies culminated in the development of a dual-FSM GI LiDAR tracking and imaging system, which, upon thorough experimental validation, demonstrated significant improvements. Notably, it achieved an improvement in the coarse tracking accuracy from 193.29 μrad (3σ) to 87.21 μrad (3σ) and enhanced the tracking accuracy from 10.1 μrad (σ) to 1.5 μrad (σ) under specified operational parameters. Furthermore, the method notably diminished the overshoot during the target capture process from 28.85% to 12.8%, concurrently facilitating clear recognition of the target contour. This research contributes significantly to the advancement of GI LiDAR technology for practical application, showcasing the potential of the proposed control and design strategies in enhancing system performance in the face of complex disturbances.

**Keywords:** GI LiDAR; flexible load; active disturbance rejection control; dual-FSM tracking and aiming; remote-sensing imaging




## 1. Introduction

Ghost imaging (GI) represents an innovative imaging paradigm that executes a correspondence operation between the detected signal light, which conveys physical light intensity information, and the measurement matrix associated with the reference light path. This technique capitalizes on the second-order intensity correlation between these two light paths to acquire the image information of the object, as delineated in the references [1–3]. Gi is different from traditional imaging methods. (1) GI can break through the limitations of traditional imaging techniques and achieve super-resolution imaging through the properties of quantum entanglement [4]. (2) GI utilizes the non-local properties

of quantum entanglement, allowing for imaging without directly illuminating the sample. This stealth imaging technology has broad application prospects in military, intelligence, and other fields [5]. (3) GI utilizes the non-local properties of quantum entanglement to achieve remote imaging. Even if the imaged object is separated from the detector by a certain distance, high-resolution images can still be obtained, which is of great significance for detection in deep sea, space, and other fields [6,7]. (4) GI can utilize the property of multi-photon entanglement to achieve multi-photon imaging, thereby obtaining more detailed image information [8,9]. Therefore, GI has advantages such as strong noise resistance and diverse imaging methods.

In 2015, the experimental equipment of the US Army Laboratory achieved laser correlation imaging from 2.33 km away [10]. In 2018, a research team from Shanghai Jiao Tong University proposed and validated fast first-photon correlation imaging based on the idea of first-photon imaging through the study of first-photon laser correlation imaging under single-photon detection mode [11]. In 2020, the research team achieved first-photon 3D correlation imaging of cooperative targets at a distance of 100 km under the condition of detecting an average of 0.01 photons per pixel [12]. In 2021, Liu Weitao's research group at the University of National Defense Technology applied the idea of pulse compression to multi-pulse time-domain correlation, achieving correlated imaging of targets at a distance of 1.3 km under conditions where the background noise of sunlight is much stronger than the target signal [13]. However, to utilize the advantages and characteristics of GI LiDAR in practical applications, key technical challenges, such as removing target motion blur and improving system environmental adaptability, still need to be overcome [14,15].

In the current GI LiDAR system, only through multiple measurements can moving targets be captured in a large field of view gaze imaging mode. Consequently, the dynamics of relative motion between the LiDAR system and the targeted object may lead to a reduction in the imaging resolution of the radar, manifesting as motion blur, as evidenced in references [15,16]. For GI LiDAR systems, traditional imaging methods that use motion compensation or fast sampling to reduce motion blur have a weaker impact on the phenomenon of motion blur. Therefore, feasible methods for removing motion blur need to be studied in conjunction with the characteristics of quantum correlation imaging.

Considering that the target movement can be categorized into two distinct components—one perpendicular to the radar's line of sight and the other parallel to it—the Shanghai Institute of Optics and Fine Mechanics at the Chinese Academy of Sciences has been at the forefront of proposing an innovative image correlation reconstruction method. This method is specifically designed to counteract the blurring of images resulting from target movements either perpendicular to or along the radar line of sight. This approach represents a significant advancement in enhancing the clarity and resolution of images captured by radar systems, addressing the challenges posed by the dynamic nature of target movement with respect to the radar's observational perspective. Among them, when the target moves perpendicular to the radar line of sight direction, a reconstruction calculation method for removing image blur is proposed for uniform speed targets using illumination light field velocity search and translation compensation [17]. When the target moves in the same direction as the radar line of sight, a reconstruction calculation method for removing image blur is proposed for uniform velocity targets using illumination light field velocity search compensation and scaling compensation [18]. The above methods have all achieved experimental verification of long-distance and high-resolution imaging, but the imaging effect is still not good for high-speed moving targets, especially variable speed targets, due to large compensation errors.

In addressing the challenge of motion blur, the research team from the Shanghai Institute of Optics and Fine Mechanics at the Chinese Academy of Sciences developed an innovative radar system for imaging moving targets utilizing single fast steering mirror (FSM) tracking. This system was successfully tested in airborne flight, achieving spatial and range resolutions surpassing 0.5 m at altitudes exceeding 1 km. Despite these advancements, the method encounters limitations due to the system's aperture receiving constraints.

Furthermore, nonlinear disturbances arising from flexible mechanisms and varied external conditions continue to pose challenges, resulting in residual motion blur during the tracking of moving targets, as detailed in the literature [3].

In response to the above situation, in order to completely solve the problem of motion blur, it is necessary to conduct research on the characteristics of GI LiDAR combined with photoelectric tracking control systems. Due to the laser wavelength emitted by the GI LiDAR system being 1064 nm, with a constraint of an emission aperture of approximately 70 mm, the existing GI LiDAR imaging resolution can be obtained to be approximately 15 µrad. Consequently, to effectively address the issue of motion blur, it is imperative that the electro-optical tracking control system achieves a tracking accuracy exceeding 5 µrad (σ) for targets moving at high speeds.

The composite axis tracking system is the main mechanism to ensure stable beam pointing and tracking [19–23]. Long-range and high-speed target capture, as well as stable imaging, are all achieved by pointing and tracking (PAT) systems. Therefore, the PAT system is integral to the functionality and effectiveness of the GI LiDAR system. The transmitting and receiving optical path is the main mechanism in the PAT system to ensure the receiving aperture and control the stability of the beam. It is located inside the optoelectronic tracking system and achieves the functions of aiming and stable imaging by controlling the stable direction of the beam. Therefore, the precision with which the transmitting and receiving optical paths are aligned directly influences the quality of imaging for high-speed targets. [24].

Elevating tracking accuracy to match the resolution of imagery necessitates overcoming substantial technical challenges, notably high-precision target tracking and the mitigation of various nonlinear disturbances, which remain critical concerns for composite axis tracking control systems. Furthermore, within the aerospace domain, there are stringent demands for peak power consumption and holding torque, necessitating the use of flexible joints, such as harmonic reduction mechanisms, to fulfill these requirements. The perturbations introduced by flexible loads profoundly affect motion control characteristics, potentially inducing mechanical resonance and precipitating system instability. Such dynamics critically undermine speed stability and positional tracking precision, presenting significant obstacles to maintaining system integrity and performance.

Therefore, there are always disturbance problems in composite axis tracking control systems. These complex disturbances include motor torque fluctuations, flexible load torque fluctuations and contact friction, platform vibration, drive system noise, etc., which affect system stability and reduce tracking accuracy. The difficulties in suppressing these complex disturbances are as follows. Firstly, the sensor's ability to collect many disturbances is limited, and the ability of traditional linear controllers to suppress nonlinear disturbances, such as motor torque and flexible loads, is significantly limited by bandwidth. Secondly, constructing a precise digital model of disturbances presents a significant challenge, hindering the controller's ability to accurately compensate for such disturbances.

In practical applications of GI LiDAR systems, the stability and accuracy of the platform's line of sight are significantly impacted by severe nonlinear disturbances, including wind resistance and line winding. Compared with linear active disturbance rejection control (ADRC), nonlinear ADRC (NLADRC) exhibits superior benefits, such as broad applicability, enhanced tracking precision, and robust resistance to interference, making it more adept at addressing these nonlinear challenges. However, its adjustable parameters are too many and difficult to adjust, which greatly limits its engineering application. Scholars have attempted to switch between linear ADRC and nonlinear ADRC to achieve better dynamic performance and anti-interference ability, but it is relatively difficult to implement, and there are still challenges to practical application [25].

In this study, aiming to address the limitations associated with the system's receiving aperture and the motion blur issues inherent in the tracking mode of the single pendulum mirror, we embarked on a collaborative research initiative with the National University of Defense Technology. This collaboration led to the proposition of a novel dual-fast steering

mirror (FSM) tracking and aiming control strategy, grounded in the principles of GI LiDAR. Firstly, our study introduces a solution to counteract the impact of nonlinear disturbances arising from both flexible mechanisms and diverse external environmental factors on motion control systems. We propose an NLADRC technique, which incorporates artificial neural networks to significantly enhance the velocity stability and tracking accuracy of GI LiDAR in real applications. This method enables the real-time optimization of parameters, effectively addressing the challenges of parameter tuning in NLADRC strategies, thereby improving the system's resistance to interference and its control precision. Secondly, in light of the imaging principles of GI and the limitations posed by the system's receiving aperture, we introduce an optical design strategy for a tracking and imaging system that employs a dual-FSM GI LiDAR. This approach substantially elevates the GI LiDAR's capability to detect high-speed moving targets. Finally, leveraging the method outlined in this manuscript, a dual-FSM GI LiDAR tracking and imaging system was developed. Therefore, the research presented herein plays a crucial role in advancing GI LiDAR technology toward practical application and technology transfer, marking a significant contribution to the field.

This paper proceeds as follows. Section 2 presents the NLADRC technique tailored for two-dimensional turntables within flexible mechanisms, along with an exploration of the primary elements influencing its precision. Section 3 offers an in-depth examination of the NLADRC strategy, particularly its implementation utilizing radial basis function (RBF) neural networks, complemented by discussions on the construction of simulation models and the analysis of resulting data. Section 4 delineates the methodology for designing the optical path of a dual-FSM tracking and imaging system, grounded in the principles of GI LiDAR. In Section 5, the development of the experimental platform for the dual-FSM tracking and imaging system is described, detailing the procedures and outcomes of tracking accuracy and imaging experiments based on GI LiDAR technology. The paper concludes with Section 6, summarizing the key findings and contributions of this research.

## 2. Related Work

Within the realm of control methodologies, the evolution of optoelectronic tracking control technology has transitioned through several stages, beginning with single-loop control; advancing to dual-loop control, encompassing both speed and position loops; and culminating in multi-loop control, which incorporates acceleration feedback. Linear feedback control methods still dominate the mainstream [26]. Although feedforward control is theoretically considered the most convenient method for optimizing control systems, its effectiveness is not ideal due to the inability to model accurately. In the field of nonlinear control, certain research results have been achieved in recent years for optoelectronic tracking systems. The control method based on the model observer has been applied in composite axis tracking control systems both theoretically and experimentally [27]. The results have confirmed that this method effectively improves the performance of the composite axis tracking control system [28]. In addition, advanced control methods, such as sliding mode control, ADRC control, and backstepping control, have gradually been applied in optoelectronic tracking control systems. As artificial intelligence continues to advance, the exploration of intelligent control technologies within the domain of composite axis tracking is progressively unfolding [25]. The composite axis tracking system can divide disturbances into internal disturbances and external disturbances according to their different sources.

Internal disturbances are usually caused by changes in system structure and parameter uncertainty, mainly including periodic disturbance torque caused by flexible joints, which can easily cause mechanical resonance, motor torque fluctuations, line disturbance torque, nonlinear friction torque between shaft systems, and periodic disturbance torque caused by mass imbalance. The impact of nonlinear disturbances caused by flexible joints and the system itself on control accuracy can be addressed by studying high-precision control methods under various nonlinear disturbances in flexible mechanisms. External disturbances

mainly come from changes in the environment in which the system operates, including platform vibrations, disturbances caused by wind resistance, and random torque impacts. Faced with these types of external disturbances, in addition to researching advanced control methods for various nonlinear disturbances, it is also necessary to study high-precision control methods for fast mirrors. This article mainly studies advanced control methods for achieving high stability and high-precision tracking under high dynamics. At present, the control methods of composite axis tracking control systems can be roughly divided into several categories: sliding film control machine, robust control, internal model control, fractional calculus control, PID control, and ADRC control [29,30].

Yuan et al. proposed a method for designing sliding mode controllers, which have advantages such as disturbance insensitivity and strong adaptability to model parameters. However, the output of the sliding mode controller may exhibit high-frequency vibrations due to the use of the sign function. In devices with high precision requirements, such as composite axis tracking control systems, such high-frequency vibrations may lead to component damage [29]. Zhu et al. proposed an H∞ robust control method based on research in the field of robust control, which balances performance indicators while improving the stability of the system. However, owing to its intricate architecture and substantial computational demands, coupled with the limitations arising from less-than-ideal strategies within robust control, this controller exhibits a conservative nature, rendering it less effective for the high-precision management of optoelectronic tracking platforms [25]. Ka et al. introduced a technique that employs a cascade extended state observer, enabling the utilization of observed errors as novel disturbances for subsequent re-estimation and compensation [31]. In the same year, Chen Jie et al. reduced the burden of the extended state observer (ESO) by collecting disturbance information at the load and motor ends and inputting it into a series elastic actuator, enabling the system to accurately estimate disturbance information while suppressing measurement noise under low bandwidth conditions [32]. Nevertheless, the controlled entities within the previously discussed active disturbance rejection controllers operate on a direct transmission basis. The presence of flexible loads significantly influences the motion control characteristics, predisposing the system to mechanical resonance, which can lead to instability. Additionally, external disturbances, primarily friction disturbances, are the main reasons for instability in optoelectronic tracking systems, especially at low speeds, which greatly affect speed smoothness and position tracking accuracy.

The essence of existing nonlinear active disturbance rejection controllers is "eliminating deviation based on deviation". Initially, this approach amalgamates both the intrinsic modeling inaccuracies of the system's controlled entity and the external perturbations, identifying them through an observer and subsequently compensating for them within the error feedback mechanism of the control rate. This ensures that the controlled object operates in alignment with the intended objectives, notwithstanding the presence of inherent model uncertainties or external disturbances. NLADRC shares a structural resemblance with its linear counterpart and primarily comprises three components: a tracking differentiator, an extended state observer, and a disturbance compensation mechanism through error feedback control. By way of illustration, the structural framework of a second-order NLADRC is depicted in Figure 1.

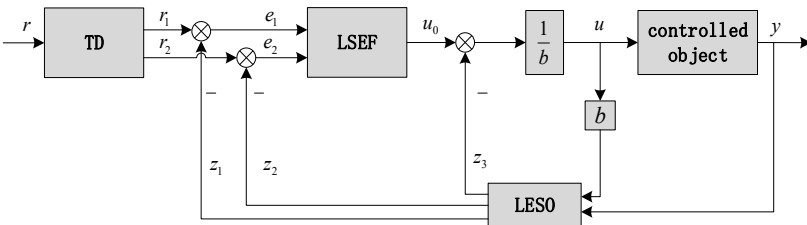

**Figure 1.** Structural diagram of second-order NLADRC.

### 2.1. Tracking Differentiator (TD)

The tracking differentiator has two output signals and one input signal at the external interface. When the controlled system operates in an environment with internal and external disturbances, the TD can output the smoothed input signal required by the system's target curve and provide an approximate differential signal. This paper first studies the second-order TD [31].

The second-order differential equations are listed:

$$\begin{cases} \dot{z}_1 = \dot{z}_2 \\ \dot{z}_2 = f(z_1, z_2) \end{cases} \tag{1}$$

All solutions are assumed to be bounded and satisfy:

$$\begin{cases} \lim\limits_{t \to \infty} z_1(t) = 0 \\ \lim\limits_{t \to \infty} z_2(t) = 0 \end{cases} \tag{2}$$

Consequently, for any bounded measurable signal $v(t)$, with $t$ in the interval [0, +∞), and for any $T > 0$, it is possible to deduce the ensuing differential equations.

$$\begin{cases} \dot{x}_1 = x_2 \\ \dot{x}_2 = r^2 f(x_1 - v(t), \frac{x_2}{r}) \end{cases} \tag{3}$$

The first component $x_1(r, t)$ of the equation solution will satisfy:

$$\lim\limits_{r \to \infty} \int_0^T |x_1(r, t) - v(t)| dt = 0 \tag{4}$$

Equation (3) can be regarded as the tracking differential equation of Equation (1). As the parameter r increases, the solution of Equation (3) can approximate $v(t)$, so it can be regarded as the differential signal of $v(t)$. However, because the continuity of the function cannot be determined, it is limited to having a solution only in the sense of Filippov. By treating $v(t)$ and its generalized functions, the generalized derivative approximating $v(t)$ can be obtained.

Utilizing the approach of the fastest discrete TD extracted from swift systems enables the circumvention of overshoot in the differential signal upon the system attaining equilibrium. Furthermore, this method eradicates the high-frequency oscillation typically observed in the differential signal output by the TD.

$$\begin{cases} fh = fhan(x_1(k) - v(k), x_2(k), r, h_0) \\ x_1(k+1) = x_1(k) + hx_2(k) \\ x_2(k+1) = x_2(k) + hfh \end{cases} \tag{5}$$

The second-order TD discretization expression of the system is shown in Equation (5).

In this paper, $x_1$ signifies the tracking signal corresponding to the input signal $v$, while $x_2$ denotes the derivative of $x_1$. The symbol $h$ represents the system's sampling period, also referred to as the integration step. By diminishing the integration step, the system's susceptibility to noise can be substantially mitigated. The variable $r$ is identified as the speed factor, influencing the system's tracking velocity, and $\beta$ is the filtering factor parameter. With a fixed value of $h$, configuring $\beta$ to exceed $h$ ensures that the derivative signal exhibits no overshoot, concurrently facilitating the effective limitation of interference within the derivative signal [32].

Equation (6) in the manuscript delineates the expression for the system's fastest control synthesis function, as discretized within this study.

$$
\begin{aligned}
&d = rh_0^2, \ a_0 = h_0 x_2 \\
&y = x_1 + a_0 \\
&a_1 = \sqrt{d(d + 8|y|)} \\
&a_2 = a_0 + \frac{sign(y)(a_1 - d)}{2} \\
&fsg(y, d) = \frac{(sign(y+d) - sign(y-d))}{2} \\
&fsg(a, d) = \frac{(sign(a+d) - sign(a-d))}{2} \\
&a = (a_0 + y - a_2)fsg(y, d) + a_2 \\
&fhan = -r\left(\frac{a}{d} - sign(a)\right)fsg(a, d) - rsign(a)
\end{aligned}
\tag{6}
$$

### 2.2. Nonlinear Extended State Observer (NESO)

In this study, a nonlinear extended state observer (ESO) is utilized to dynamically monitor and predict both internal and external disturbances affecting the system. Serving as the cornerstone of ADRC, the ESO not only facilitates real-time tracking of the system's output variables but also enables the estimation of emergent disturbance signals by analyzing the behavior of the controlled object. This analysis allows for the incorporation of these disturbances into the system's state variables, thereby enhancing the control rate module's error feedback mechanism. Remarkably, the ESO's estimation capabilities depend solely on the input and output data of the controlled object, compensating for discrepancies through the disturbance variables observed [33]. This approach obviates the need for precise mathematical modeling of the controlled object and its disturbance signals, showcasing the method's exceptional robustness.

Firstly, the second-order object is modeled and analyzed, and the state space equations are listed, as shown in Equation (7):

$$
\begin{cases}
\dot{x}_1 = x_2 \\
\dot{x}_2 = f(x_1, x_2, t) + \omega(t) + bu \\
y = x_1
\end{cases}
\tag{7}
$$

Then, the internal disturbance $f(x_1, x_2, t)$ and the external disturbance $w(t)$ are expanded to a new total disturbance state variable, denoted as $w_a(t)$, where:

$$
\begin{cases}
\dot{x}_1 = x_2 \\
\dot{x}_2 = x_3 + bu \\
\dot{x}_3 = \dot{\omega}_a(t) = \omega_0(t) \\
x_1 = y
\end{cases}
\tag{8}
$$

In response, a nonlinear state observer for the system is conceptualized, as delineated in Equation (9):

$$
\begin{cases}
e = z_1 - y \\
\dot{z}_1 = z_2 - \beta_1 e \\
\dot{z}_2 = z_3 - \beta_2 \cdot fal(e, a_1, \delta) + bu \\
\dot{z}_3 = -\beta_3 \cdot fal(e, a_2, \delta)
\end{cases}
\tag{9}
$$

In the formula:
$z_1, z_2$—State variables observed by the observer;
$z_3$—Estimated total disturbance;
$\beta$—Observer gain coefficient;
$e$—Error between the observed variable and the output variable;
$a$—Factors of non-linear segment intervals;

$\delta$—Length of the nonlinear interval, $0 < \delta <$ one;

$b$—Compensation coefficient.

After discretization, it becomes:

$$\begin{cases} \varepsilon_1(k) = z_1(k) - y(k) \\ z_1(k+1) = z_1(k) + h[z_2(k) - \beta_1\varepsilon_1] \\ z_2(k+1) = z_2(k) + h[z_3(k) - \beta_2 fal(\varepsilon_1(k), a_1, \delta_1) + bu] \\ z_3(k+1) = z_3(k) - h\beta_3 fal(\varepsilon_1(k), a_2, \delta_1) \end{cases} \tag{10}$$

Within this framework, $h$ represents the size of the integration step, while $\beta_1$, $\beta_2$, $\beta_3$, $a_1$, $a_2$, and $a_3$ are tunable parameters. The *fal* function is identified as a distinctive nonlinear function that exhibits the output characteristic of "large error, small gain" and "small error, large gain". In other words, a smaller gain coefficient is adopted to reduce overshoot when the error is relatively large, and a larger gain coefficient is used to improve the system's speed when the error is small [34]. The specific expression is:

$$fal(e, a, \delta) = \begin{cases} \frac{e}{\delta^{a-1}} &, |e| \leq \delta \\ |e|^a sign(e), & |e| > \delta \end{cases} \tag{11}$$

Derived from Equation (10), it becomes evident that upon the determination of the controller parameters, the observer is capable of monitoring the real-time state variables of the system along with the augmented disturbance variables. This enables compensation within the control rate, ultimately facilitating the estimation of the disturbance state variables.

### 2.3. Nonlinear State Error Feedback Control Rate (NLSEF)

Within the framework of a closed-loop system, the configuration of the error feedback rate plays a pivotal role in determining the efficacy of the control system. Traditional PID controllers typically employ a straightforward linear aggregation of the error signal, alongside its integral and differential counterparts, to endow the controller with rudimentary corrective capabilities. Conversely, the nonlinear state error feedback (NLSEF) mechanism integral to ADRC utilizes specialized nonlinear functions for weighting the error signal, thereby enhancing efficiency beyond what is achievable with linear PID control strategies. The ESO within ADRC meticulously monitors the system's aggregate disturbance variable, encompassing both the internal modeling bias disturbance and external perturbations. By leveraging precise error feedback protocols within the control rate, the control variable of the system is ascertained, facilitating effective disturbance compensation within the control system [34].

In ADRC, the main function of NLSEF is to fit the state error and compensate for disturbances. To eliminate errors as much as possible and obtain error feedback control quantities, we adopt the following nonlinear combination:

$$u_0 = \beta_4 \cdot fal(e_1, a_3, \delta) + \beta_5 \cdot fal(e_2, a_4, \delta) \tag{12}$$

In Equation (12), $e_1$ and $e_2$ are the errors between the tracking signals $x_1$ and $x_2$ put out by the tracking differentiator and the state variables $z_1$ and $z_2$ observed by the observer, respectively. The discrete form of NLSEF is shown in Formula (13), which eliminates disturbance interference on the system by feeding back the disturbance state variable $z_3$ estimated by ESO, where b is the system gain, and the gain coefficients $\beta_4$, $\beta_5$, $\delta_2$, $a_3$, and $a_4$

are all parameters to be tuned. The methods for tuning these parameters will be elaborated upon in subsequent sections of this text.

$$
\begin{cases}
e_1(k) = x_1(k) - z_1(k) \\
e_2(k) = x_2(k) - z_2(k) \\
u_0(k) = \beta_4 \cdot fal(e_1(k), a_3, \delta) + \beta_5 \cdot fal(e_2(k), a_4, \delta) \\
u(k) = \frac{u_0(k) - z_3(k)}{b}
\end{cases}
\tag{13}
$$

In adherence to the foundational principles of ADRC, the methodology for controller modeling entails the dynamic observation of state and disturbance variables, coupled with the processing of error feedback. Firstly, the transition mechanism of the tracking differentiator is employed to derive the input state variables of the system along with their differential signals, aiming to preempt overshoot in the command signal and enhance the promptness with which state variables align with command signals. Subsequently, the extended state observer is tasked with monitoring the actual system's real-time output state of the controlled object, enabling the estimation of both internal modeling bias disturbance and external interference, thereby facilitating the observation of expanded state variables. Finally, a designated error feedback mechanism within the control rate is utilized to ascertain the control variable of the controlled object, accomplishing disturbance compensation within the control system. According to ADRC's composition principles, this approach offers a more rapid response to reference commands than traditional PID controllers, eliminates overshoot during operation, and ensures a swift return to the original static working point in the face of severe nonlinear disturbances, such as wind resistance and line winding, all while maintaining minimal steady-state errors.

Existing nonlinear ADRCs consist of three parts, each of which has parameters that need to be tuned. The combination of sizes of each parameter leads to different effects, making it more difficult to adjust them together. Traditional second-order nonlinear ADRC has as many as 11 parameters to be tuned, which consumes a lot of time and effort. Therefore, to leverage the advantages of NLADRC, it is necessary to design suitable parameter-tuning methods. Existing parameter-tuning methods can be roughly divided into four categories: separation method, index optimization method, structure optimization method, and artificial intelligence method.

## 3. Nonlinear ADRC Method Control Method Based on RBF Neural Network

In this segment, we explore the superiority of NLADRC over its linear counterpart (LADRC), noting its broader applicability, enhanced tracking precision, and superior disturbance negation prowess. To tackle the intricate challenge of parameter tuning within NLADRC, a novel approach utilizing a radial basis function neural network identifier (RBFNNI) for real-time parameter adjustment is introduced. This method capitalizes on the inherent self-learning and adaptability features of neural networks. Specifically, the RBFNNI is deployed for real-time identification within a dual-position loop NLADRC system, which integrates harmonic speed reducers, thereby facilitating the acquisition of dynamic information about the controlled object. Building on this, a parameter tuning strategy predicated on the RBFNNI for the NLADRC controller is developed. This strategy is aimed at enabling the real-time modification of parameters within the nonlinear error feedback mechanism of the controller, thus significantly enhancing the dynamic response and disturbance mitigation efficacy of the turntable rough tracking system.

### *3.1. Radial Basis Function Neural Network*

Radial basis function networks can be divided into two types: regularized networks and generalized networks. Among them, the number of hidden nodes in the regularized networks needs to be strictly consistent with the number of input training samples, which results in regularized networks calculating weights in practical applications $\omega_{ij}$; it is neces-

sary to calculate the inverse of the $N \times N$ matrix with a complexity of approximately $o(N^3)$. That is, as $N$ increases, the likelihood of a sick matrix increases, and the computational complexity rapidly increases. Therefore, in order for regularized networks to achieve higher accuracy and better performance, a sufficient number of samples must be given, which leads to huge computational complexity, low efficiency, and difficulty in implementation.

The generalized network can be obtained by regularizing the network changes. The structure of the generalized radial basis function network used in this article is shown in Figure 2:

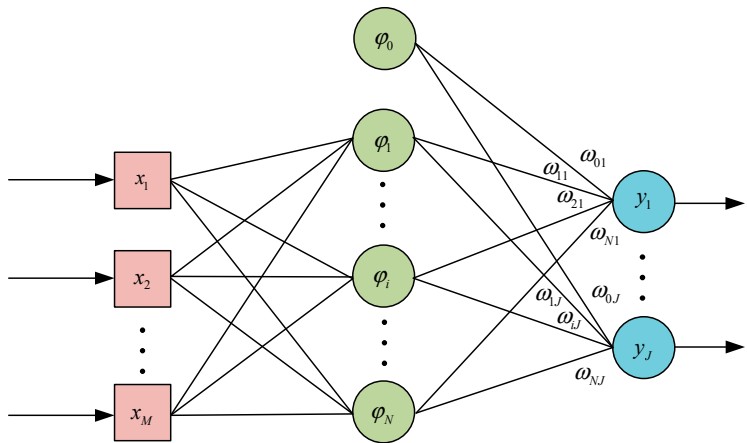

**Figure 2.** Generalized radial basis function neural network structure.

Mirroring the structure of regularized networks, generalized radial basis function (RBF) networks feature a three-layer architecture, inclusive of an input layer equipped with $M$ nodes. The distinguishing factor lies in the specification that the count of nodes $I$ within the hidden layer must not exceed the number of training samples. Furthermore, the basis function associated with each hidden node $i$ is identified as $\phi(\|X - X_i\|)$, with $X_i = [x_{i1}, x_{i2}, \cdots, x_{im}]$ representing the center of this basis function. Also, a separate threshold $\Phi_0$ is added and connected to the output node, and the weight is represented as $\omega_{0j}$. The output layer also has J output nodes.

$Y_k = [y_{k1}, y_{k2}, \cdots, y_{kj}, \cdots y_{kJ}]$ is used to represent the actual output and $k$ represent the input vector index. The result of the $j$th output node of the generalized network for the training sample $X_k$ can be expressed as:

$$y_{kj} = \omega_{0j} + \sum_{i=1}^{I} \omega_{ij} \phi(X_k, X_i), j = 1, 2, \cdots, J \tag{14}$$

This paper uses the Green function as the basis function, i.e.,:

$$\phi(X_k, X_j) = G(X_k, X_j) \tag{15}$$

Therefore, $\phi(X_k, X_j)$ can be expressed as:

$$\phi(X_k, X_j) = G(X_k, X_j) = G(\|X_k - X_j\|) = \exp\left(-\frac{1}{2\sigma^2}\|X_k - X_j\|\right) \tag{16}$$

### 3.2. Nonlinear Active Disturbance Rejection Controller Based on RBF Neural Network

Existing second-order NLADRCs have 11 parameters to be tuned, significantly increasing the tuning difficulty and hindering the application of ADRC methods in aerospace systems. In NLADRC, the control quantity is obtained through nonlinear combinations of error feedback. Given the pronounced influence of the parameters $\beta_4$ and $\beta_5$ on the efficacy of the nonlinear state error feedback control rate within controllers, this chapter leverages the self-learning and adaptability inherent in neural networks. These networks

are harnessed to perform identification tasks within a dual-position loop ADRC system that incorporates harmonic speed reducers, facilitating the acquisition of real-time dynamic information about the controlled entity. Building on this foundation, a parameter optimization approach utilizing RBF neural networks is developed for the ADRC controller. This method enables the real-time refinement of the $\beta_4$ and $\beta_5$ parameters within the nonlinear error feedback mechanism, significantly improving the dynamic responsiveness and disturbance mitigation capacity of the turntable rough tracking system.

Figure 3 illustrates the schematic representation of the NLADRC system, which is underpinned by an RBF neural network.

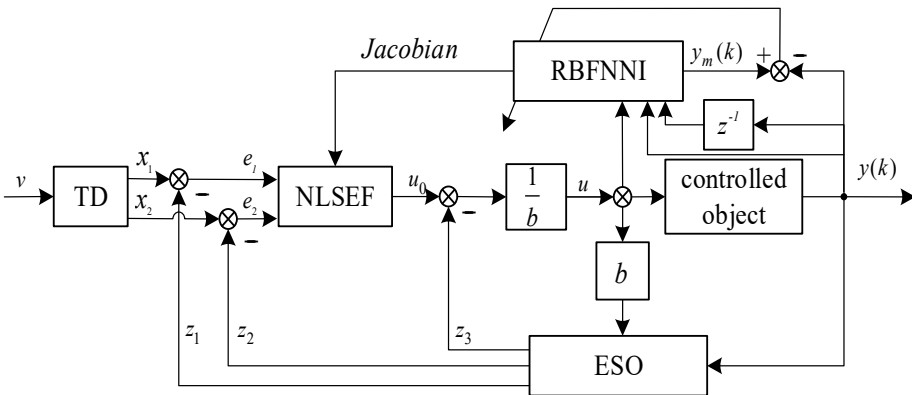

**Figure 3.** Nonlinear ADRC control structure diagram based on RBF neural network.

The structure diagram mainly consists of three parts: the NLADRC controller, the radial basis function neural network identifier (RBFNNI), and the controlled object. The composition of the NLADRC has been described in detail above. The input of the network identifier consists of the controlled quantity received by the controlled object and positional information, and the online identification of the identifier optimizes the parameters inside the neural network in real time. This ensures that the network output continuously approaches the feedback position of the actual controlled object. Simultaneously, the sensitivity information of the controlled object's output response to input changes (Jacobian information) is calculated and injected into the nonlinear active disturbance rejection controller's NLSEF, ultimately achieving online adjustment of $\beta_4$, $\beta_5$ parameters in the controller's nonlinear error feedback combination.

The identification method of the RBF neural network can be classified into offline identification and online identification on the basis of different real-time characteristics. Offline identification requires identification before the system operates, using previously collected system input-output information to train the network. This method does not consider real-time changes in the system, mainly used for predicting fixed models, resulting in fast identification speed but lacking real-time performance. Online identification, on the other hand, is conducted during the operation of the control system. It utilizes real-time updates of training samples based on input and output signals, using fitting errors to correct network parameters in real time, thereby achieving synchronous identification of the system. This study chose online identification. Initially, the structure and number of nodes of the RBF neural network are determined, and the parameters of the NLADRC based on RBF neural networks are initialized. Subsequently, the control input and feedback velocity of the controlled object are fed into the identifier, whereupon the neural network undergoes training, and the parameters within the network are adjusted in real time. This process facilitates the real-time tuning of the NLADRC parameters, thereby enhancing the precision of the rough tracking system's control mechanism.

The foundational architecture of the RBF neural network has previously been outlined. The performance index function selected for this discussion is represented as follows in Equation (17):

$$E(k) = \frac{1}{2}(y(k) - y_m(k))^2 \tag{17}$$

where $y(k)$ denotes the system's actual speed feedback and $y_m(k)$ is the speed output predicted by the RBF neural network. The adjustment of the network parameters, including the center vector, basis width vector, and weight vector of the hidden layer, is essential for system optimization. The iterative algorithm employed in this study for parameter optimization is the gradient descent method, with the specific computational formula provided in Equation (18):

$$\begin{cases} \Delta c_{ji}(k) = \frac{\partial E(k)}{\partial c_{ji}(k)} = (y(k) - y_m(k))\omega_j(k)\frac{x_i(k) - c_{ji}(k)}{b_j(k)^2} \\ c_{ji}(k) = c_{ji}(k-1) + \eta_1 \Delta c_{ji}(k) + \alpha\big(c_{ji}(k-1) - c_{ji}(k-2)\big) \end{cases}$$

$$\begin{cases} \Delta b_j(k) = \frac{\partial E(k)}{\partial b_j k} = (y(k) - y_m(k))\omega_j(k)h_j(k)\frac{\|X(k) - C_j(k)\|^2}{b_j(k)^3} \\ b_j(k) = b_j(k-1) + \eta_2 \Delta b_j(k) + \alpha\big(b_j(k-1) - b_j(k-2)\big) \end{cases} \tag{18}$$

$$\begin{cases} \Delta \omega_j(k) = \frac{\partial E(k)}{\partial \omega_j k} = (y(k) - y_m(k))h_j(k) \\ \omega_j(k) = \omega_j(k-1) + \eta_3 \Delta \omega_j(k) + \alpha\big(\omega_j(k-1) - \omega_j(k-2)\big) \end{cases}$$

where $X = [u(k), y(k), y(k-1)]^T$ constitutes the input vector for the RBF neural network, $\eta_1$ is the learning rate, and $\alpha$ represents the momentum factor.

After obtaining the identification object's fitting output, the radial basis function (RBF) neural network is capable of acquiring a real-time dynamic measure, specifically, the partial derivative of the output relative to the input, which is the Jacobian information as delineated in the subsequent equation. As analyzed earlier, the neural network, after being trained, will obtain an identification output $y_m(k)$ with an extremely small identification error. On the basis of this, it can be considered that the identification output $y_m(k)$ is equivalent to the actual output $y(k)$, and thus we have:

$$\frac{\partial y(k)}{\partial u(k)} \approx \frac{\partial y_m(k)}{\partial u(k)} = \sum_{j=1}^{m} \omega_j(k)h_j(k)\frac{c_j(k) - u(k)}{b_j(k)^2} \tag{19}$$

When performing real-time tuning of the control rate module parameter $\beta$ in the NLADRC controller in the system, we select a performance index function, which can be expressed as follows:

$$E_1(k) = \frac{1}{2}(r(k) - y(k))^2 \tag{20}$$

where $r(k)$ denotes the system's input signal and $y(k)$ signifies the system's output signal. In this paper, the adjustment of $\beta_4$ and $\beta_5$ is conducted using the gradient descent method, as shown in the following formula:

$$\begin{cases} \Delta \beta_4(k) = -\eta\frac{\partial E_1(k)}{\partial \beta_4(k)} = -\eta\frac{\partial E_1(k)}{\partial y(k)}\frac{\partial y(k)}{\partial u(k)}\frac{\partial u(k)}{\partial \beta(k)} \\ \qquad = \eta(r(k) - y(k))\frac{\partial y(k)}{\partial u(k)}fal(e_1, a_1, \delta) \\ \beta_4(k) = \beta_4(k-1) + \Delta \beta_4(k) \\ \Delta \beta_5(k) = -\eta\frac{\partial E_1(k)}{\partial \beta_5(k)} = -\eta\frac{\partial E_1(k)}{\partial y(k)}\frac{\partial y(k)}{\partial u(k)}\frac{\partial u(k)}{\partial \beta_5(k)} \\ \qquad = \eta(r(k) - y(k))\frac{\partial y(k)}{\partial u(k)}fal(e_2, a_2, \delta) \\ \beta_5(k) = \beta_5(k-1) + \Delta \beta_5(k) \end{cases} \tag{21}$$

In the formula, $\frac{\partial y(k)}{\partial u(k)}$ represents the Jacobian information, as shown in Equation (21). $\eta$ represents the learning rate, and it satisfies $0 < \eta < 1$.

The above is the neural network parameter tuning scheme designed in this paper. From Equation (21), it can be seen that obtaining accurate partial derivative information is crucial in the parameter update algorithm. Some scholars have used the sign function or sigmoid function to compute this partial derivative information, but both may introduce significant errors, impacting control accuracy. In contemporary system identification efforts, leveraging Jacobian information as demonstrated in Equation (21)—representing the output's sensitivity to input variations and offering the optimal linear approximation of a differentiable equation at a specific point—has become prevalent.

### 3.3. Simulation Verification of Nonlinear Active Disturbance Rejection Control Method via RBF Neural Network

To comprehensively assess the performance attributes of the control methodology proposed in this study, the models of LADRC, Lagrange LADRC, and the nonlinear ADRC approach underpinned by the RBF neural network, as discussed in the preceding sections, are methodically evaluated within the MATLAB simulation environment. By comparison with similar methods, the anti-disturbance ability, speed, and position-tracking performance of the proposed RBF-based nonlinear active disturbance rejection control method are verified. The built simulation model is shown in Figure 4.

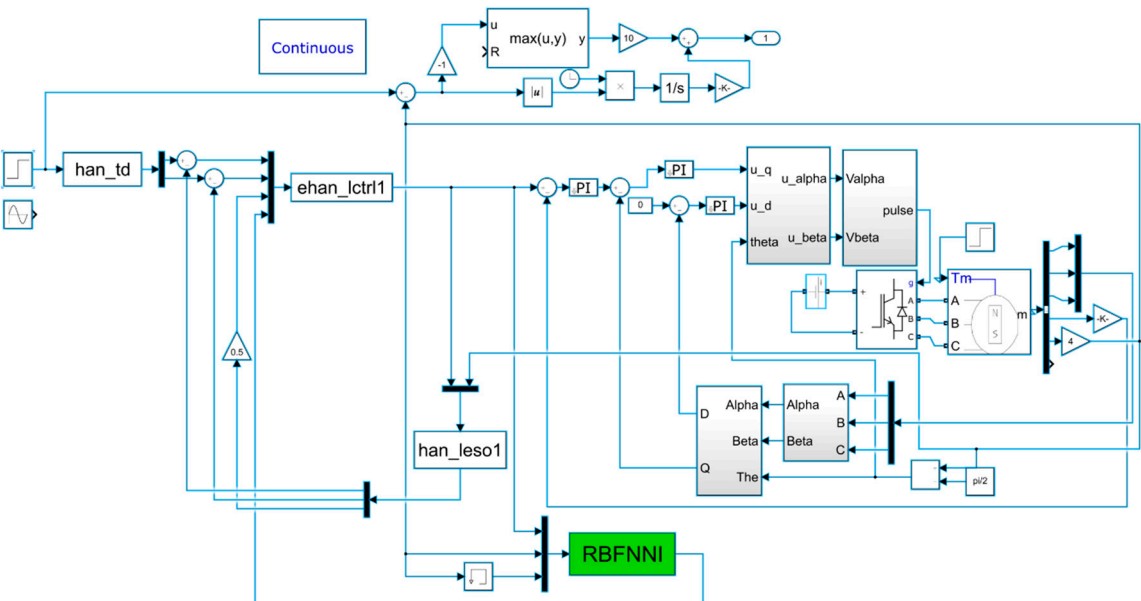

**Figure 4.** Simulation diagram of nonlinear ADRC control based on RBF neural network.

The parameter values used in the existing NLADRC for the turntable are as follows:

Parameters in the TD module: $r_0 = 10$, $h_0 = 0.01$; parameters in the ESO: $\delta_1 = 0.05$, $a_1 = 0.5$, $a_2 = 0.25$, $b = 1.5$; $\beta_1 = 3\omega_0$, $\beta_2 = 3\omega_0^2$, $\beta_3 = \omega_0^3$, where $\omega_0$ represents the observer bandwidth; parameters in the NLSEF: $\delta_2 = 0.05$, $a_3 = 0.75$, $a_4 = 1.25$, $\beta_4 = 1.84$, $\beta_5 = 2.76$.

In the implementation of the RBF neural network within the framework of NLADRC for the turntable system, the parameters of the NLADRC are largely retained. The RBF neural network identifier is structured with three input nodes in the initial layer, five nodes in the intermediary hidden layer, and two output nodes in the terminal layer. The designated learning rate for the RBF neural network is set at $\eta_1 = 0.3$ with a momentum factor of $\alpha = 0.032$. For the state error feedback control component of the NLADRC, the learning rate is established at $\eta = 0.15$.

(a)    Velocity Step Response Simulation Analysis

To evaluate the efficacy of the proposed control method in mitigating disturbances, a simulation analysis focusing on the velocity step response is executed under two distinct scenarios: in the absence of disturbance and in the presence of disturbance. When the setpoint curve is a step function with a desired velocity of $1°/s$, the simulation results for the three controllers under the condition without disturbance are shown in Figure 5.

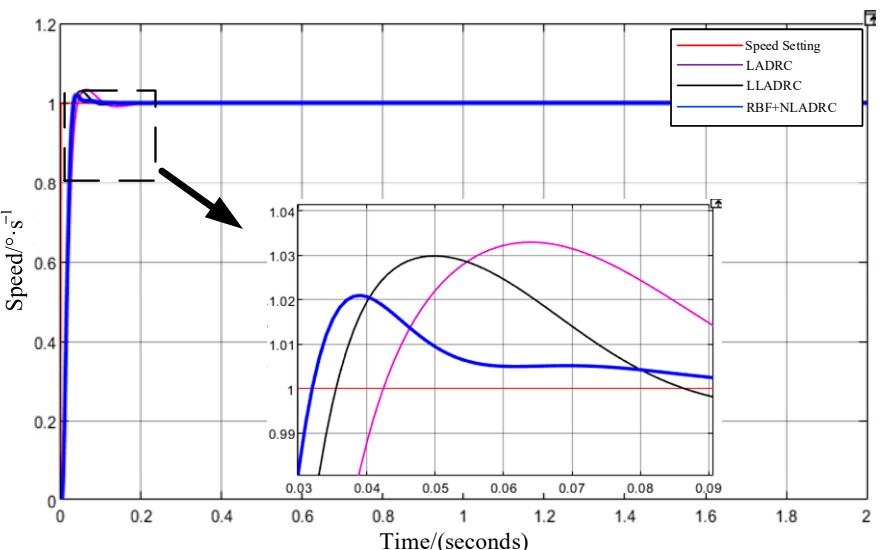

**Figure 5.** Simulation comparison of speed step response.

The variation process of parameter tuning for the NLADRC controller via the RBF neural network during the operation of the turntable is shown in Figure 6.

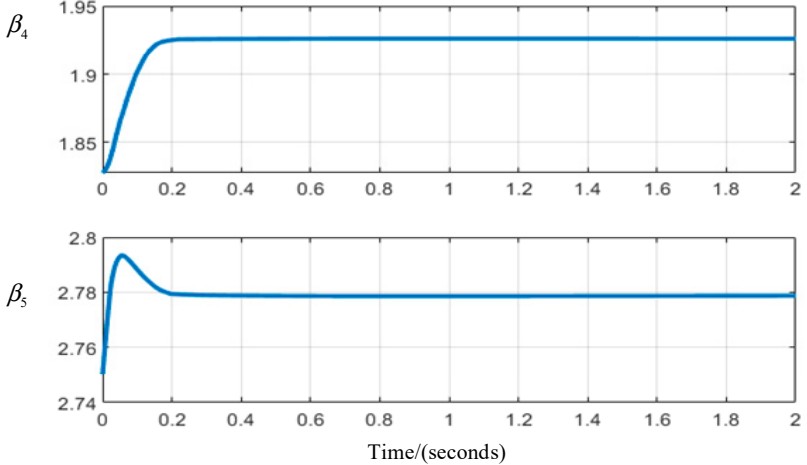

**Figure 6.** Parameter variation curve (without disturbance).

On the basis of Figure 6 and Table 1, it can be concluded that the three control methods exhibit minimal settling time and overshoot for speed stability control in the absence of external disturbances. Nevertheless, the simulation outcomes reveal that the response velocity of the RBF-based nonlinear active disturbance rejection control approach is superior. Compared with the Lagrange LADRC control method, the adjustment period of this technique is decreased by 0.024 s, and the overshoot is diminished by 28%.

**Table 1.** Performance indicators of speed control (without disturbance).

| Control Algorithm | Settling Time (s) | Overshoot (%) |
|---|---|---|
| LADRC | 0.085 | 3.36 |
| Lagrange-LADRC | 0.065 | 2.96 |
| RBF+NLADRC | 0.041 | 2.11 |

To assess the disturbance rejection capabilities of the control strategy delineated in this study, an external environmental perturbation, emulating wind interference through white noise, was superimposed on the system's feedback curve during a step input at a setpoint velocity of $1°/s$. This disturbance was integrated into the output angle feedback of the turntable to simulate real environmental conditions. Figure 7 subsequently presents the outcomes of velocity stability simulation experiments conducted under the three different control methodologies for comparative analysis.

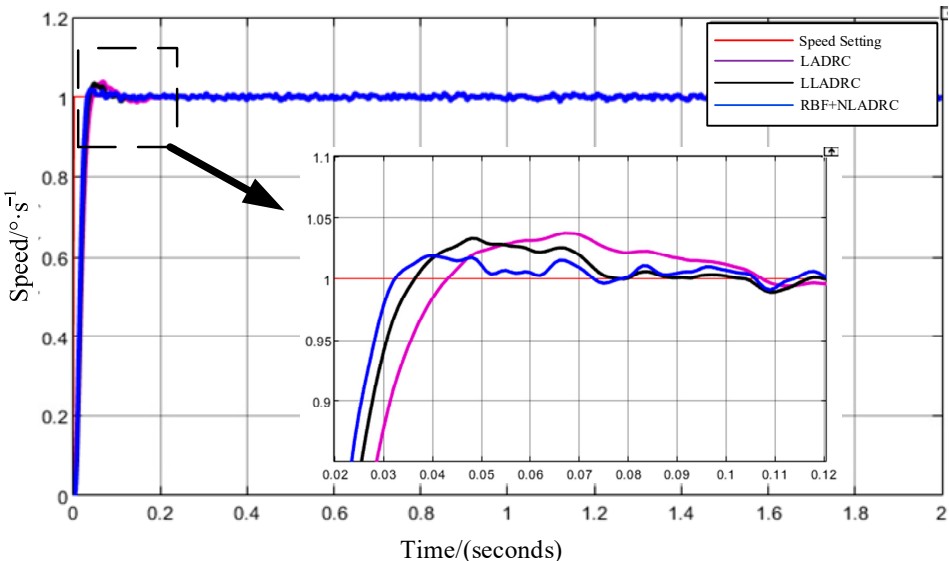

**Figure 7.** Feedback speed comparison (with disturbance).

The comparative analysis depicted in Figure 7 demonstrates that the turntable controller employing RBF-based nonlinear active disturbance rejection control achieves a reduced adjustment period and superior velocity stability in the face of external disturbances relative to the other two control strategies. This observation underscores the RBF nonlinear active disturbance rejection control method's enhanced dynamic and steady-state efficacy in facilitating coarse tracking of the two-dimensional turntable, particularly when contrasted with the Lagrange LADRC control approach.

Furthermore, by examining the variation curves of the simulation parameters in Figure 8 and Table 2 during the system control process, it can be observed that the RBF non-linear active disturbance rejection control method has a certain adaptive capability, as it can dynamically optimize controller parameters on the basis of variations in internal and external disturbances. This capability allows for improved system control precision and disturbance rejection performance.

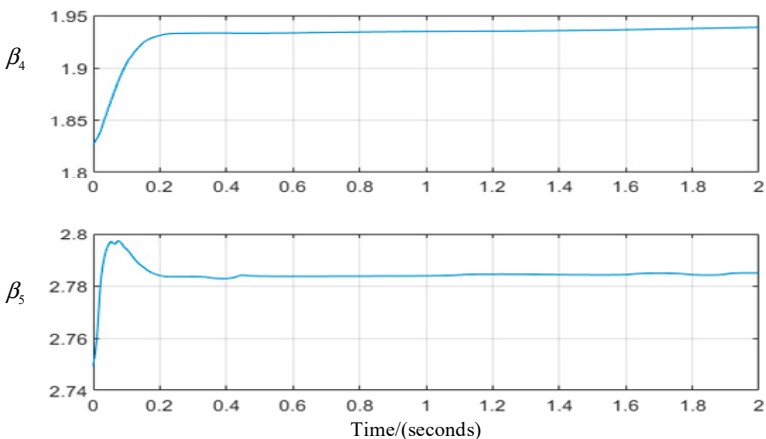

**Figure 8.** Parameter variation curve (with disturbance).

**Table 2.** Performance indicators of speed control (with disturbances).

| Control Algorithm | Speed Stability (°/s) |
| --- | --- |
| LADRC | 0.064 |
| Lagrange LADRC | 0.062 |
| RBF+NLADRC | 0.05 |

Simulation Analysis of Position Tracking Accuracy

The comparison of position feedback for different control methods when the system is given a sinusoidal target curve is shown in Figure 9.

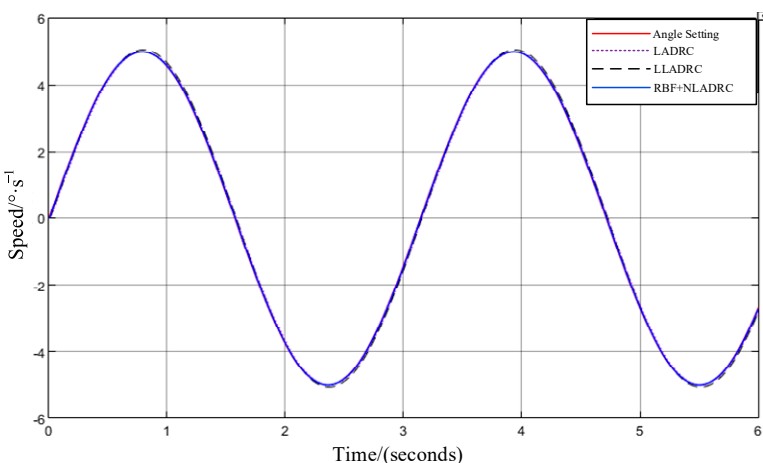

**Figure 9.** Comparison of sinusoidal target tracking position curves.

Through the comparison in Figure 10 and Table 3, it is evident that under disturbance-free conditions, the position tracking performance of the RBF nonlinear active disturbance rejection control method is superior to Lagrange LADRC.

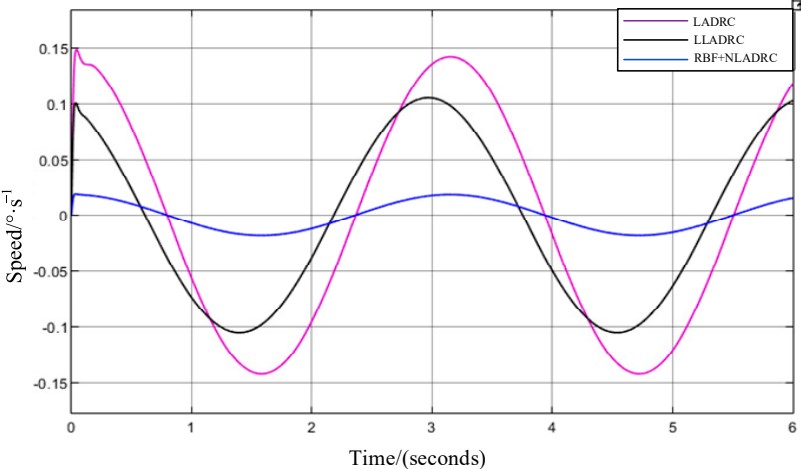

**Figure 10.** Sine position tracking error curve.

**Table 3.** Performance indicators of position control (without disturbance).

| Control Algorithm | Position Tracking Error (°) |
| --- | --- |
| LADRC | 0.1422 |
| Lagrange LADRC | 0.1052 |
| RBF+NLADRC | 0.0184 |

The curve comparison of position tracking with system disturbance introduced is presented in Figure 11.

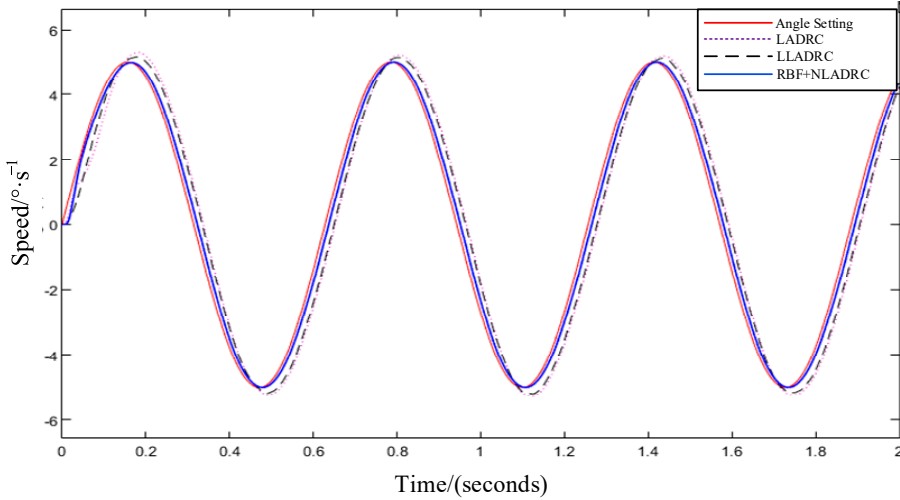

**Figure 11.** Comparison curves of position tracking for different control methods (with disturbance).

In Figure 12 and Table 4, it can be observed that after introducing disturbances into the turntable system, the error magnitude for the Lagrange LADRC control method is 0.213°, while for the NLADRC control method based on RBF, the error magnitude is 0.032°. Therefore, compared with the Lagrange LADRC control method, the NLADRC control method based on RBF achieves an 85% reduction in position tracking error, indicating superior anti-disturbance and position tracking performance.

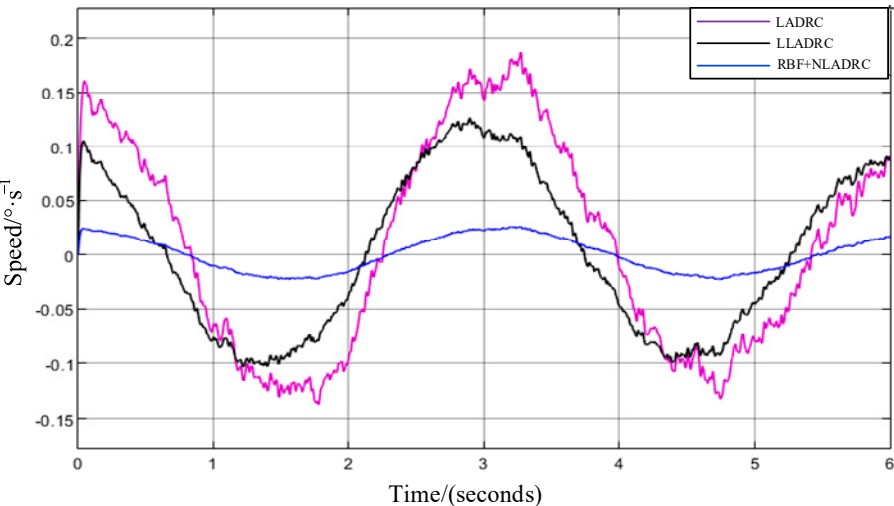

**Figure 12.** Position tracking error of turntable (with disturbance).

**Table 4.** Performance indicators for position tracking control (with disturbance).

| Control Algorithm | Position Tracking Error (°) |
| :---: | :---: |
| LADRC | 0.252 |
| Lagrange LADRC | 0.213 |
| RBF+NLADRC | 0.032 |

According to the above simulation experimental data, due to the NLADRC control method based on RBF, the parameters within the control system can be optimized in real time on the basis of variations in internal and external disturbances. This method not only achieves higher control accuracy and stronger disturbance rejection performance than similar methods but also possesses certain adaptability.

## 4. Design Approach for Dual-FSM 3D GI Tracking Imaging Control System

The optical structure diagram of a typical GI LiDAR system is depicted in Figure 13 below. This system is bifurcated into a laser emission subsystem and a laser tracking subsystem. In the laser transmitting system, a pulsed spatiotemporal two-dimensional random speckle field is generated through a pulsed pseudo-thermal light source plane. Subsequently, the beam is divided into two separate beams via a beam splitter. Among them, the reflected beam illuminates the pseudo-thermal light source plane's speckle pattern onto a CCD through a reference mirror, capturing the spatial intensity distribution of the illumination field. The transmitted beam uses a transmitting mirror to project the illuminated speckle pattern onto the target.

Within the laser tracking system, the dispersed echo signal from the target is concentrated onto a single-pixel photomultiplier tube (PMT) via a receiving mirror. This signal is then captured as a sequence of voltage signals by a high-speed acquisition card located in a processing computer. Associating these signals with the spatial intensity distribution of the speckle pattern, as recorded by the laser transmitting system, enables the GI LiDAR system to reconstruct the correlation image. Consequently, this process yields an intensity distribution image of the target's surface, as observed through the capabilities of the GI LiDAR system.

The operational principles of laser three-dimensional intensity correlated imaging radar reveal that, firstly, by employing spatial intensity coding, the radar system is capable of capturing high-dimensional target information using point detectors. This approach not only minimizes the requirement for extensive detection apparatus but also improves the system's resistance to interference within intricate channel conditions. Secondly, the

paradigm of information gathering, predicated on second-order correlation, necessitates multiple samplings. Consequently, this methodology intrinsically encounters a motion blur challenge when tracking moving targets [21].

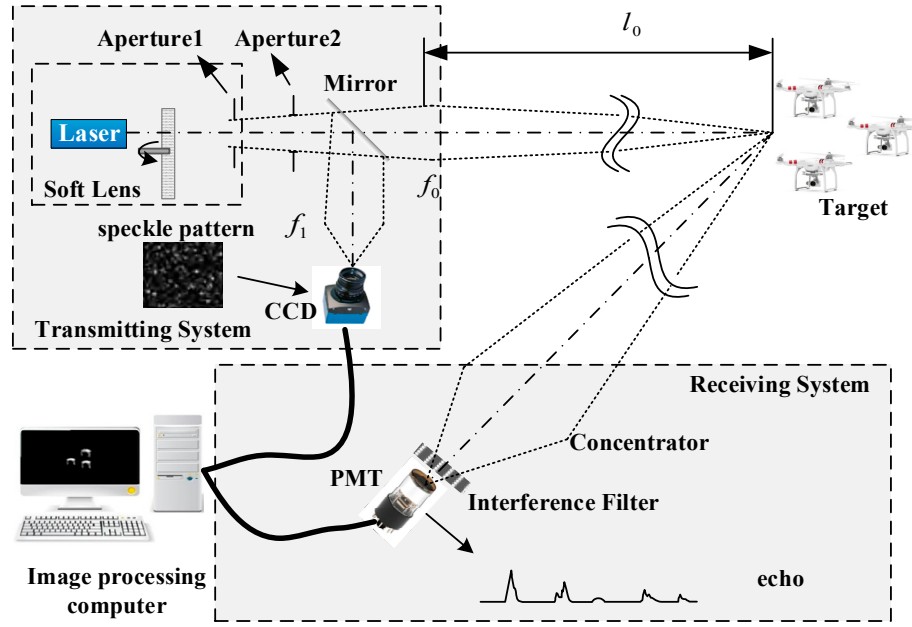

**Figure 13.** Typical optical structure diagram of GI LiDAR.

Addressing the issues of system receiving aperture limitations and motion blur in the tracking mechanism of the single pendulum mirror, our collaboration with the Shanghai Institute of Optics and Fine Mechanics at the Chinese Academy of Sciences has led to the development of a moving-target GI LiDAR system, utilizing a dual-FSM tracking approach, as depicted in Figure 14. This innovative system is structured into two main components: a transmission system and a reception system, detailed in Figures 14a and 14b, respectively.

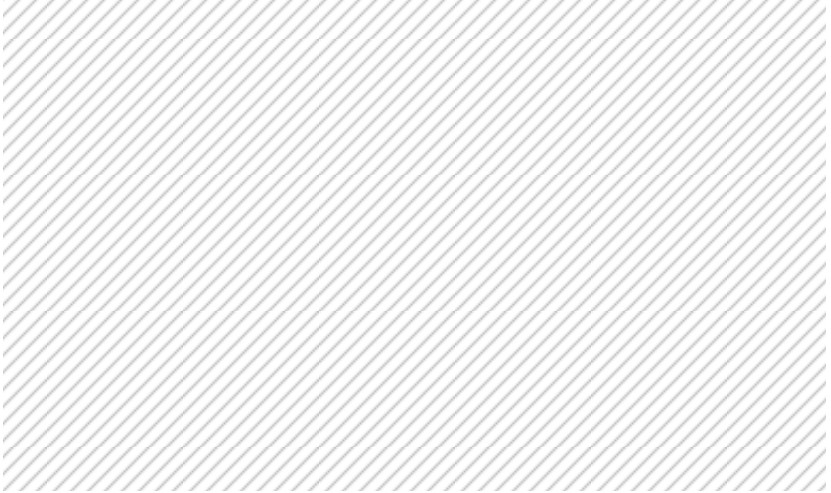

**Figure 14.** *Cont.*

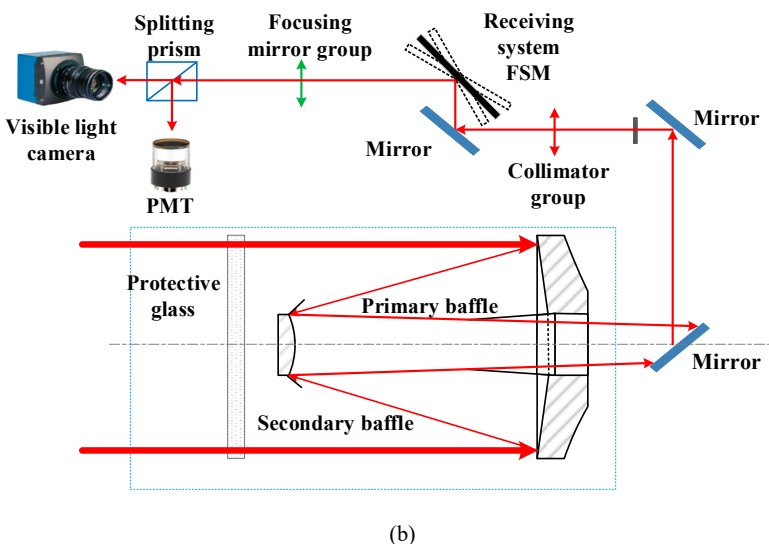

(b)

**Figure 14.** Schematic representation of the experimental setup for the dual-FSM GI LiDAR tracking control system (PMT: photon multiplier tube). (**a**) Optical path configuration of the transmitting system; (**b**) optical path configuration of the tracking system.

### 4.1. Transmitting System

To enhance the receiving aperture of the GI LiDAR system, this study introduces a GI LiDAR configuration utilizing a dual-FSM approach, advancing beyond the capabilities of the conventional single FSM setup. The developed dual-FSM GI LiDAR composite tracking optical system integrates a laser transmitting system with a laser tracking system, synergistically facilitating the projection of the laser onto the target surface and enabling stable, high-precision tracking of the target.

Figure 14 illustrates an innovative optical system delineated in this manuscript, which marries off-axis laser transmission with dual-FSM and dual-band composite tracking for cohesive imaging. The transmission architecture encompasses a reference optical path module, an object optical path transmission module, and a linkage tracking module. Specifically, as depicted in Figure 14a, the laser emission mechanism employs a converging lens to focus a 60 mm aperture parallel laser beam optimally. This focused beam undergoes encoding at a coding plate, after which a beam splitter divides it into two distinct paths. The first path projects onto a reference camera through a reference imaging mirror, serving as the reference light, while the second path, constituting the object light path, is directed toward the target via the transmission mirror assembly, inclusive of the FSM within the transmitting system. The active tracking module in the receiving system calculates the deflection angle of the FSM on the basis of the optical magnification ratio, ensuring that the dual-FSM across both laser tracking and transmitting systems collaboratively facilitates the stable tracking of moving targets and the achievement of laser correlation imaging. The efficacy of tracking directly influences the echo signal's intensity and the resultant laser correlation imaging's resolution.

### 4.2. Tracking System

The tracking system comprises the object light path reception module and the active tracking module. As shown in Figure 14b, the proposed laser tracking system in this paper is a dual-band (visible light and 1064 nm laser) common aperture optical system. The visible light system is a compound tracking system designed in a modular way. Firstly, the echo energy from the target is received by the Cassegrain main system and then collimated and converged into ideal parallel light through the collimating lens assembly. The parallel light is then directed onto the FSM in the tracking system. The tracking FSM is located in the parallel light path, ensuring that it maintains ideal imaging while in operation. Secondly,

the parallel light beam is deflected by the FSM and converges onto the visible light camera and photomultiplier tube (PMT) for imaging, using imaging mirror assemblies and a beam splitter, respectively. Finally, high-precision compound tracking is achieved through the FSM. The FSM in the transmitting system is coordinated with the FSM in the tracking system, ensuring continuous alignment between the transmitting and tracking ends. This ensures that the coding laser emitted by the transmitting end is well imaged onto the center of the PMT, and the coding correlation imaging is achieved by matching it image with the reference camera of the transmitting end.

Within the tracking subsystem, target-originated echo signals are captured by the Cassegrain receiving collimating mirror and subsequently relayed to the PMT via the FSM dedicated to tracking. This particular FSM functions as the active tracking component. The active tracking module is designed to operate within the spectral range of 480 nm to 650 nm for efficient target tracking. Photons reflected from the target falling within this spectral window are projected onto the monitoring CCD camera. The imagery captured by this camera undergoes analysis by the tracking mechanism to determine the miss distance, the results of which are then fed into the control system. By manipulating the composite axis tracking facilitated by dual-FSM, the system achieves real-time laser correlation imaging.

## 5. Experimental Platform and Result Analysis

The dual-FSM GI LiDAR system, as investigated in this study, is primarily engineered for spatial search, target identification, and subsequent lock-on and tracking operations. In alignment with the optical system's architectural blueprint, the system's payload is categorized into two main segments: the laser transmitting system and the laser tracking system. Within the laser tracking system, both the composite control unit and the video processing unit are integrated, culminating in a streamlined dual-FSM GI LiDAR imaging configuration.

### 5.1. Laser Transmitting System Experimental Platform

Figure 15 presents the experimental platform's configuration for the dual-FSM GI LiDAR laser transmitting system. This system encompasses three key segments: the reference light path module, the object light path emission module, and the integrated tracking module.

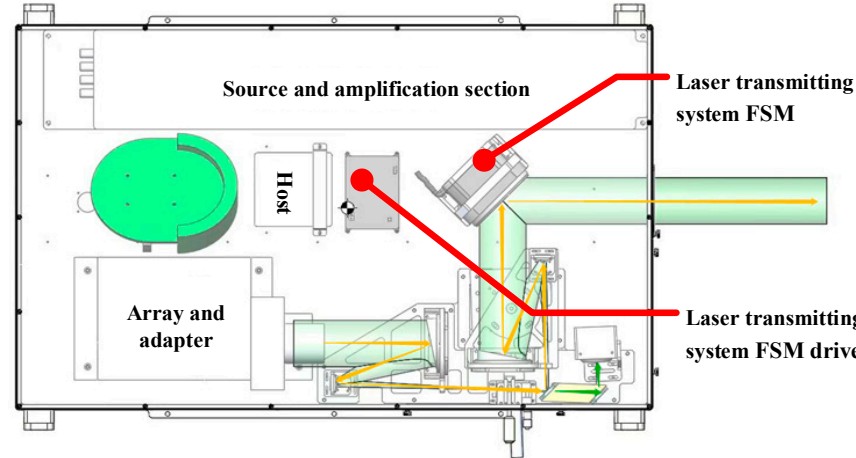

**Figure 15.** Optical system composition experimental platform model of Dual-FSM GI LiDAR laser transmitting system.

### 5.2. Laser Tracking System Experimental Platform

Figure 16 illustrates the model of the experimental platform for the dual-FSM GI LiDAR laser reception system. This system is comprised of two main components: the object light path reception module and the active tracking module.

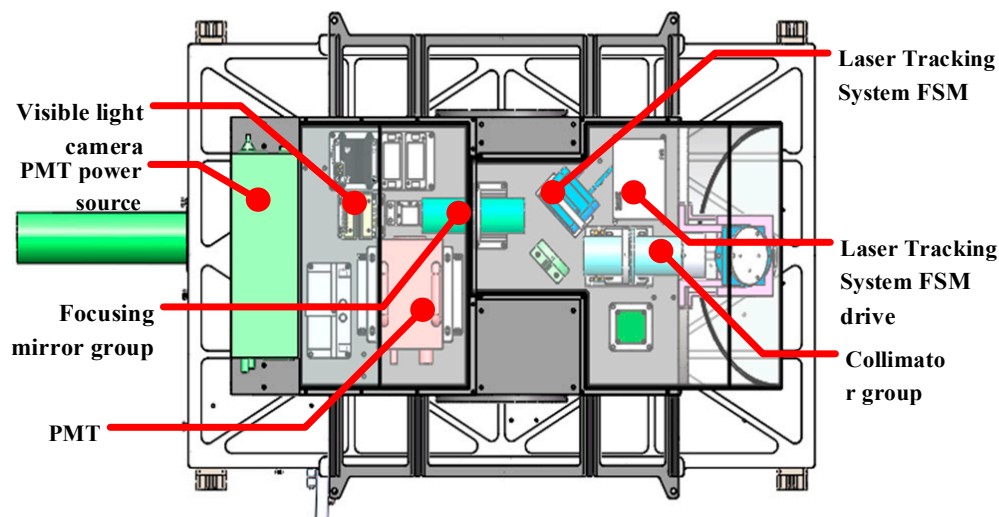

**Figure 16.** Optical system composition experimental platform model of dual-FSM GI LiDAR laser tracking system.

Figure 17 illustrates the three-dimensional physical model of the dual-FSM GI Li-DAR tracking and imaging system. The overall system model is shown in Figure 17a, which, from a structural perspective, consists of the main system components: a two-dimensional turntable, a laser transmitting system, and a laser tracking system. The two-dimensional turntable is composed of azimuth and elevation axes, each equipped with a motor, a harmonic reducer, angle measurement components at the motor and load ends, and limit mechanisms.

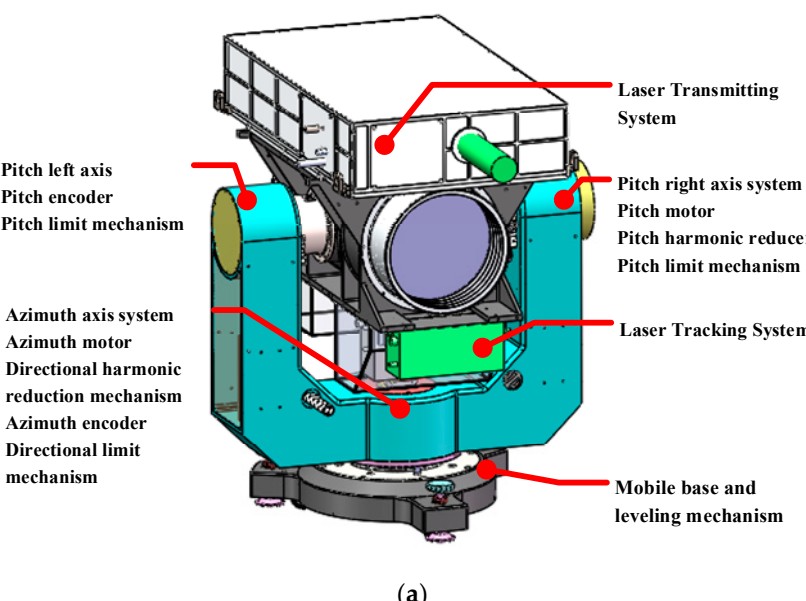

(**a**)

**Figure 17.** *Cont.*

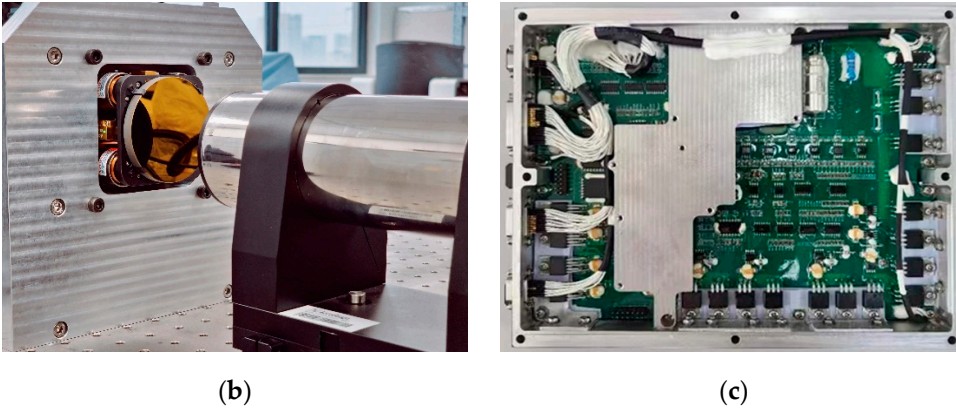

**(b)**               **(c)**

**Figure 17.** Photo of experimental platform: (**a**) 3D Physical Model of dual-FSM GI LiDAR tracking imaging system; (**b**) physical image of FSM for laser tracking system; (**c**) physical image of two-dimensional turntable composite axis controller.

Figure 17b shows the physical model of the FSM of the laser reception system, while Figure 17c shows the physical model of the composite shaft controller for the two-dimensional turntable.

### 5.3. The Experimental Results of the Target Tracking Accuracy of the Dual-FSM GI LiDAR

During the outdoor tracking experiment involving the dual-FSM GI LiDAR system, the designated target was a DJI Phantom 4 unmanned aerial vehicle (UAV), measuring 289.5 mm in each dimension. The UAV was positioned 2.74 km away from the GI LiDAR imaging apparatus, with its maximum velocity reaching up to 7 m/s. To thoroughly evaluate the efficacy of the previously introduced technological advancements, the study segmented the target tracking experiment into two distinct phases: an initial test focusing on coarse tracking precision, followed by an assessment of composite axis tracking accuracy.

#### 5.3.1. Accuracy Testing of dual-FSM GI LiDAR Coarse Tracking UAV

In order to verify the accuracy comparison of different control methods under flexible loads and various interferences, a harmonic reduction mechanism with a reduction ratio of 80:1 was used as the transmission mechanism in the two-dimensional turntable of the experimental platform.

We conducted comparative experiments on the coarse tracking accuracy of existing LADRC control methods, Lagrange LADRC control methods, and the proposed NLADRC method based on RBF neural networks under flexible transmission mechanisms and various interferences. The outcomes of the experiments are depicted in Figure 18 and detailed in Table 5, respectively. Comparative experiments were conducted on the coarse tracking accuracy of existing LADRC control methods, Lagrange LADRC control methods, and NLADRC control methods based on RBF neural networks under flexible transmission mechanisms and various interferences. The outcomes of the experiments are depicted in Figure 19 and detailed in Table 5, respectively.

**Table 5.** Comparison of coarse tracking accuracy results of three methods.

| Control Algorithm | Overshoot (%) | Tracking Error (μrad) |
|---|---|---|
| LADRC | 28.85 | 193.29 |
| Lagrange-LADRC | 14.4 | 187.17 |
| RBF+NLADRC | 12.8 | 87.21 |

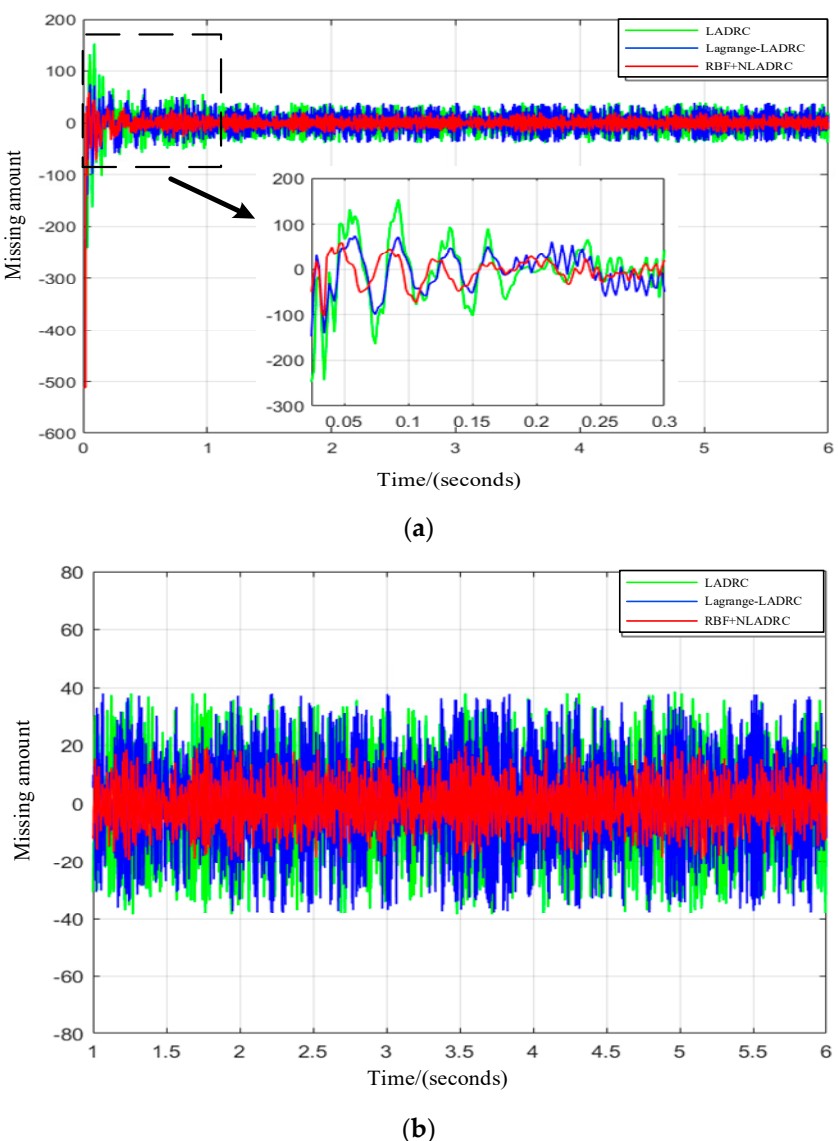

**Figure 18.** Comparative experiment on coarse tracking accuracy of two-dimensional turntable under different control methods with UAV under different speeds. (**a**) LADRC control method; (**b**) NLADRC control method via RBF neural network.

The data gleaned from the conducted experiments indicate that the conventional LADRC control technique is prone to the highest level of overshoot in the target acquisition phase, peaking at 28.85%, alongside a tracking discrepancy of 193.29 µrad (3σ). Conversely, both the Lagrange LADRC and the RBF neural network-enhanced NLADRC control strategies report a comparable overshoot rate of around 12.8% in the target capture scenario. Regarding the precision of target tracking, the RBF neural network-supported NLADRC approach outperforms in terms of anti-interference strength and reaction velocity, registering a tracking precision of 87.21 µrad (3σ). These outcomes corroborate the efficacy of the control methodologies explored within this study.

The comparative analysis of the preceding experiments demonstrates that the NLADRC control strategy, underpinned by an RBF neural network, exhibits superior anti-interference capabilities and achieves the highest level of tracking precision.

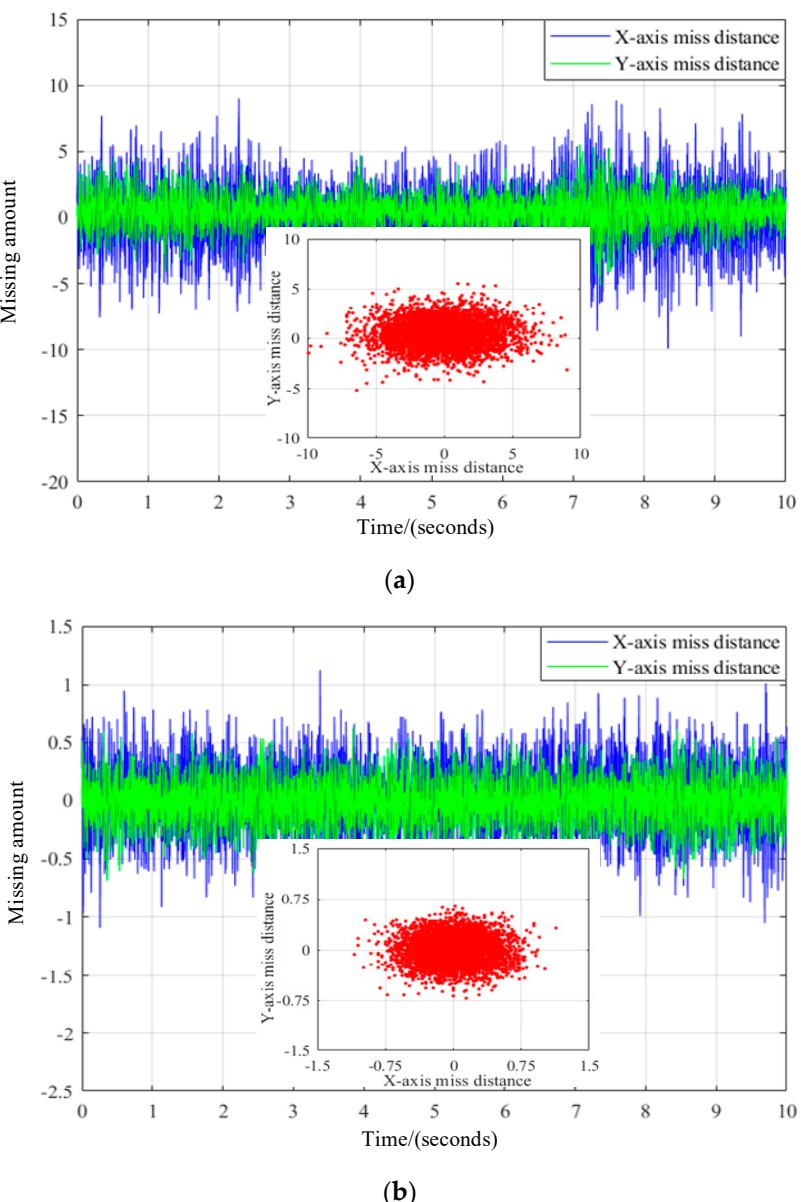

**Figure 19.** Comparison experiment of dual-FSM GI LiDAR composite axis tracking accuracy under different control methods with UAV under different speeds. (**a**) LADRC control method; (**b**) NLADRC control method based on RBF neural network.

5.3.2. Tracking Accuracy Test of Dual-FSM GI LiDAR Composite Axis Tracking UAV

To assess the impact of various control strategies on the tracking precision of composite axes in UAV tracking, experimental evaluations were performed utilizing both the traditional LADRC control technique and the innovative NLADRC method, which incorporates an RBF neural network. The outcomes of these tests, including tracking error trajectories for each control method, are presented in Figure 19a,b, and detailed data are presented in Table 6. Given the angular resolution requirement for miss distance at 5.1 μrad (σ), the tracking accuracy achieved with conventional algorithms yielded 18.8 μrad (σ) along the *X*-axis and 10.1 μrad (σ) along the *Y*-axis. By contrast, the accuracy obtained through the newly proposed NLADRC method, as depicted in Figure 19b, significantly surpasses these figures, recording an *X*-axis accuracy of 2.2 μrad (σ) and a *Y*-axis accuracy of 1.5 μrad (σ), thereby aligning with the resolution specifications necessary for GI LiDAR imaging.

**Table 6.** Comparison of composite tracking accuracy results between two methods.

| Control Algorithm | Coordinate Axis | Tracking Error ($\mu$rad) |
|---|---|---|
| LADRC | x | 18.8 |
| | y | 10.1 |
| RBF+NLADRC | x | 2.2 |
| | y | 1.5 |

### 5.4. GI LiDAR Tracking UAV IMAGING Test

To evaluate the dual-FSM GI LiDAR tracking imaging system and assess the influence of the NLADRC control method, which employs an RBF neural network, on the system's imaging capabilities in comparison to conventional control strategies, field tests were conducted on the tracking and imaging of a UAV. In addition, during system tracking, the acquisition frequency of the echo signal is 20 KHz, and algorithms such as differential ghost imaging (DGI), compressive sensing (CS), and the CLEAR algorithm were used for image processing. Figures 20 and 21 depict the imaging results captured by the dual-FSM GI LiDAR tracking imaging system during the UAV's outfield tracking. The tests were carried out at a distance of 2.74 km, featuring a DJI Phantom 4 UAV with dimensions of 289.5 $\times$ 289.5 $\times$ 289.5 mm. Throughout the testing phase, the UAV's flight velocities were varied between hover, 5 m/s, and 7 m/s.

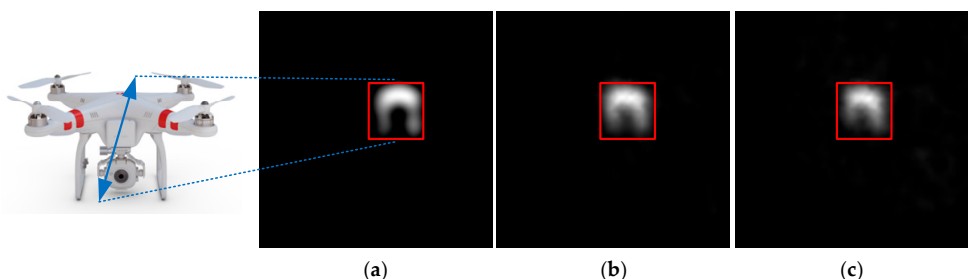

(**a**)                      (**b**)                      (**c**)

**Figure 20.** Comparison of imaging effects of dual-FSM GI LiDAR System combined with existing control algorithms with UAV under different speeds. (**a**) UAV hovering; (**b**) UAV V = 5 m/s; (**c**) UAV V = 7 m/s.

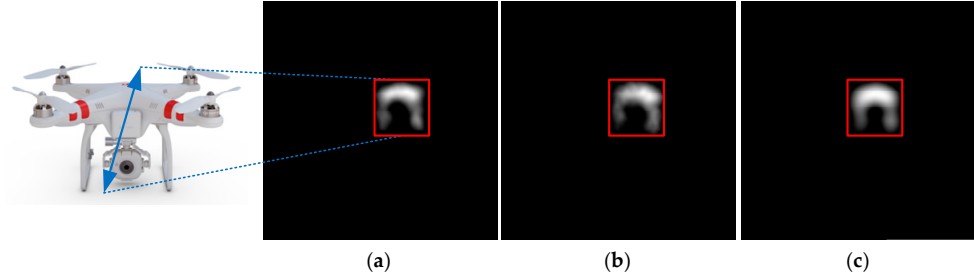

(**a**)                      (**b**)                      (**c**)

**Figure 21.** Comparison of Imaging Effects of dual-FSM GI LiDAR System combined with proposing control algorithms with UAV under different speeds. (**a**) UAV hovering; (**b**) UAV V = 5 m/s; (**c**) UAV V = 7 m/s.

Figure 20 illustrates the imaging outcomes of the UAV utilizing conventional control techniques. In scenarios in which the UAV maintains a hover, indicating minimal relative motion speed with the target, the imaging clarity of the system remains relatively high. However, as the target's velocity escalates, the resultant imaging of the drone exhibits noticeable blurring. This degradation in image quality is attributed to the existing methods' diminished tracking precision for dynamically fast-moving targets. Conversely, Figure 21

showcases the imaging performance of the UAV when subjected to the control methodology proposed in this study. While the UAV is in hover, the imaging results are comparable to those achieved with traditional methods. Nonetheless, a marked enhancement in imaging clarity is observed as the UAV accelerates, particularly at higher speeds. These findings corroborate the efficacy of the introduced tracking control strategy and the design approach of the dual-FSM GI LiDAR tracking imaging system, underscoring their superiority in capturing high-quality images of high-velocity targets.

## 6. Conclusions

This study addresses the challenges of translating GI LiDAR technology into practical applications, focusing on the dual-FSM composite axis control technique within the GI LiDAR framework. It explores the high-resolution imaging detection of high-speed moving targets across three dimensions. Initially, in response to the issues of maintaining high stability and precision in controlling two-dimensional turntables amidst nonlinear disturbances from flexible mechanisms and typical nonlinear factors, an NLADRC strategy employing an RBF neural network was introduced. This strategy notably enhanced coarse tracking accuracy from 193.29 μrad (3σ) to 87.21 μrad (3σ). Subsequently, to augment the radar's receiving aperture while simultaneously elevating tracking precision and imaging quality, a novel optical path design method for a dual-FSM GI LiDAR tracking and imaging system was proposed. The experimental outcomes affirmed the viability of this approach. Furthermore, the development of a dual-FSM GI LiDAR tracking and imaging system culminated in the achievement of a tracking accuracy of 1.5 μrad (σ). The tests underscore the system's capability for high-resolution imaging of high-speed moving targets, marking a significant advancement towards the practical application of GI LiDAR in remote sensing imaging detection. Currently, the validated target distance using this methodology extends to 2.74 km, involving a cooperative target. However, GI Lidar technology will have application requirements for tracking and detecting non-cooperative targets in the future. It is also necessary to study the target position's accurate detection method combined with the echo signal's energy in GI LiDAR so that the GI LiDAR system can play a greater role in the field of target detection.

**Author Contributions:** Conceptualization, Y.C., M.X., F.W. and H.W.; methodology, Y.C., F.W. and M.G.; software, Y.C.; validation, Y.C., H.W., K.J. and L.W.; formal analysis, Y.C.; investigation, W.H.; resources, W.H.; data curation, L.W.; writing—original draft preparation, Y.C. and L.W.; writing—review and editing, Y.C., H.W. and K.J.; visualization, S.G.; supervision, Y.C. and M.G.; project administration, M.X.; funding acquisition, W.H. and F.W. All authors have read and agreed to the published version of the manuscript.

**Funding:** This research received no external funding.

**Data Availability Statement:** The raw data supporting the conclusions of this article will be made available by the authors on request.

**Acknowledgments:** The authors express sincere thanks for the experiments provided by the Photoelectric Tracking and Measurement Technology Laboratory, Xi'an Institute of Optics and Precision Mechanics, CAS, dual-FSM two-dimensional turntable compound axis tracking and pointing project of Xi'an Institute of Optics and Precision Mechanics (Grant No. E29031G1A1), China.

**Conflicts of Interest:** The authors declare no conflicts of interest.

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
