# Peer review of "A Dual-FSM GI LiDAR Imaging Control Method Based on Two-Dimensional Flexible Turntable Composite Axis Tracking"

_remotesensing, doi:10.3390/rs16101679_

Round 1

Reviewer 1 Report

Comments and Suggestions for Authors

GI LiDAR is capable to capture the diffraction-limited images of the object, which can provide much more details than the point cloud produced by conventional LiDAR. Therefore, GI LiDAR is promising in standoff sensing. Developing the accompanied tracking and pointing system is prerequisite to put GI LiDAR in practical application. Especially, the accuracy of the tracking system has a significant influence on the spatial resolution of the image captured GI. Here, Cao et al. develop a 2-dimensional tracking system, which is achieved by a dual-FSM and turntable composite axis controlled by a disturbance rejection algorithm. They state that the accuracy of the tracking can reach up to 1.5μrad, which is impressive. The imaging results for a moving UAV are also convincing. I think this technique worth to be known by the scientific society. However, I still have some serious concerns as follows,

1.      The language in the manuscript need to be improved a lot. For example, the sentence ‘…(NLADRC) method rooted in artificial…’ in the abstract is confusing. If my interpretation is right, the expression should be revised as ‘…(NLADRC) method based on artificial…’, another example, the sentence ‘…a reduction in the coarse tracking accuracy from 193.29μrad to 87.21μrad…’ should revised as ‘…an improvement in the coarse tracking accuracy from 193.29μrad to 87.21μrad…’There are some other issues on the language. The language of the manuscript should be polished carefully.

2.      The illustrations of the figure need to be detailed. It takes me a lot of time to search between the lines for the context related to the figures, due to the missing details in the illustration. In addition, I think Figure 15 can be deleted, since there seems no difference between Fig.15 and Fig.18.

3.      There are some parameters which is missing the representation, such as ym(k) in Eq.17

4.      The image formation of Ghost imaging and more details about the experiment should be provided in the part of ‘5.4 GI LiDAR tracking UAV imaging test’, since the imaging results is reconstructed via GI method. How many samplings have been performed, what reconstruction algorithm is used? These important details should be given.

 Nevertheless, the proposed technique is important sufficiently that it warrants being returned for a revision in the hope that an improved version will measure up to being published.

Comments on the Quality of English Language

There are a lot of work on the language need to be done, to make the things clear and concise.

Author Response

Thank you very much for your attention and the valuable comments on our paper. We have revised the manuscript based on your feedback, please refer to the PDF attachment for details.

Reviewer 2 Report

Comments and Suggestions for Authors

The paper presents a sophisticated method for enhancing GI LiDAR imaging, focusing on a dual-FSM (fast steering mirror) control system. It introduces a novel control strategy and an optical design to improve tracking accuracy and system performance against nonlinear disturbances. The results show significant improvements in tracking accuracy.

Issues to be addressed:

1. The introduction section contains excessive details. For example, the relevance of mentioning the Shanghai Institute of Optics and Fine Mechanics and the Xi'an Institute of Optics and Fine Mechanics is unclear. Such details may be perceived as unrelated to the core content of this academic paper, which should not serve as marketing material to highlight others' successes.

2. The literature review appears overly focused on the works from the Chinese Academy of Sciences, suggesting a limited review of international research. This could imply a narrow scope of consultation, which might undermine the paper's comprehensive understanding of the global research landscape.

3. Section 2 includes excessive details, particularly regarding unused methods. Please refrain from listing detailed equations not directly applied in the paper. If included, these equations should be properly referenced. The focus should primarily be on the contributions of this work.

4. The paper should clearly outline the study's limitations and suggest directions for future research, providing detailed information to enable replication, including any necessary software or data.

5. Clarify the application of the technology, explaining the necessity of tracking moving UAVs and its importance. Detail the specific benefits this technology offers for remote sensing, particularly in civilian applications, to highlight its relevance and practical value.

Comments on the Quality of English Language

Moderate editing of English language required. Lots of grammar and spelling mistakes.

Author Response

Dear Reviewer 2:

Thank you very much for your attention and the valuable comments on our paper. We have revised the manuscript based on your feedback, please refer to the PDF attachment for details.

Reviewer 3 Report

Comments and Suggestions for Authors

The article presents the control system for gi lidar. The topic of the article is highly actual. The authors achieved an improvement in steering accuracy in the presence of interference, which they document in key figures 5, 7, 12 and tables 2 – 4. The resulting effect of increased sensing accuracy is documented in Fig. 21 and 22 and supported by the experimental results. However, I noticed some flaws in the article that need to be removed, sse comments. I consider the number of references to literature to be below average and I recommend the authors to significantly expand it.

Comments:

1. References should be arranged in ascending order in the text.

2. Italics should be used to indicate quantities in the text.

3. Page 4: Reference [38] does not exist.

4. Page 6: "In this paper, v signifies the tracking signal corresponding to the input signal v". Are the input and trace signal names the same (v)?

5. Equations (1) to (13) are in section 2 – Related works, but it is not clear from which references they are taken.

6. Page 12: "Where 𝑦(𝑘) denotes the system's actual speed feedback, and 𝑦(𝑘) is the speed output predicted by the RBF neural network", I think that following equation (17) the explanation of the symbol 𝑦m( 𝑘) is missing.

Author Response

Dear Reviewer 3:

Thank you very much for your attention and the valuable comments on our paper. We have revised the manuscript based on your feedback, please refer to the PDF attachment for details.

Round 2

Reviewer 2 Report

Comments and Suggestions for Authors

The authors have solved all issues. 

Reviewer 3 Report

Comments and Suggestions for Authors

The authors in the second revision addressed all of my previous comments.